# Effects of Water Depth on the Growth of the Submerged Macrophytes *Vallisneria natans* and *Hydrilla verticillata*: Implications for Water Level Management

Qisheng Li, Yanqing Han, Kunquan Chen, Xiaolong Huang ®, Kuanyi Li and Hu He *®

State Key Laboratory of Lake Science and Environment, Nanjing Institute of Geography and Limnology, Chinese Academy of Sciences, Nanjing 210008, China; liqisheng515@163.com (Q.L.); hanyanqing18@mails.ucas.ac.cn (Y.H.); chenkq816@163.com (K.C.); xlhuang@niglas.ac.cn (X.H.); kyli@niglas.ac.cn (K.L.)
* Correspondence: hehu@niglas.ac.cn; Tel.: +86-25-8688-2237

**Abstract:** Water level is one of the most important factors affecting the growth of submerged macrophytes in aquatic ecosystems. The rosette plant *Vallisneria natans* and the erect plant *Hydrilla verticillata* are two common submerged macrophytes in lakes of the middle and lower reaches of the Yangtze River, China. How water level fluctuations affect their growth and competition is still unknown. In this study, three water depths (50 cm, 150 cm, and 250 cm) were established to explore the responses in growth and competitive patterns of the two plant species to water depth under mixed planting conditions. The results show that, compared with shallow water conditions (50 cm), the growth of both submerged macrophytes was severely suppressed in deep water depth (250 cm), while only *V. natans* was inhibited under intermediate water depth (150 cm). Moreover, the ratio of biomass of *V. natans* to *H. verticillata* gradually increased with increasing water depth, indicating that deep water enhanced the competitive advantage of *V. natans* over *H.verticillata*. Morphological adaptation of the two submerged macrophytes to water depth was different. With increasing water depth, *H. verticillata* increased its height, at the cost of reduced plant numbers to adapt to poor light conditions. A similar strategy was also observed in *V. natans*, when water depth increased from 50 cm to 150 cm. However, both the plant height and number were reduced at deep water depth (250 cm). Our study suggests that water level reduction in lake restoration efforts could increase the total biomass of submerged macrophytes, but the domination of key plants, such as *V. natans*, may decrease.

**Keywords:** lake restoration; morphological response; interspecific competition; light attenuation



## 1. Introduction

In freshwater lakes, submerged macrophytes are related to various biogenic elements (N, P, S, etc.) in the water body, and they are the shapers of lake ecosystem structure and the maintainers of lake ecosystem function [1,2]. For example, studies of temperate lakes have shown that submerged macrophytes can influence the structure of food webs by providing spawning grounds and habitats for fish [3] and supplying abundant food and shelter for zooplankton and invertebrates [4]. Submerged macrophytes can also promote the settlement of suspended particulate matter in water [5] and inhibit the growth of phytoplankton through allelopathy and nutrient competition [6].

Water level is one of the main factors affecting the growth and distribution of submerged macrophytes in shallow lakes [7–10]. Within a specific range, a decrease in water level will increase the light intensity in the water, thus increasing the biomass and diversity of submerged macrophytes [11]. For instance, in Lake Okeechobee, USA, the lake depth in 2000 (2.8~3.7 m) was significantly lower than that in 1999 (4.0~5.2 m). Simultaneously, the biomass of submerged macrophytes in the lake increased rapidly from 5 g m$^{-2}$ (dry weight) in 1999 to 12 g m$^{-2}$ in 2000, and the distribution area expanded from 15 km$^2$ in

1999 to 180 km$^2$ [12]. Geest investigated 100 floodplain lakes in the lower Rhine River and found that the species richness of submerged macrophytes was significantly positively correlated with the proportion of certain water levels (the ratio of the exposed lake floor area after water depth decreased in October to the lake floor area during the high-water season in July), indicating that the decrease in water depth was conducive to the increase in submerged macrophyte diversity [13]. In lake restoration in China, decreasing water level was also often used to rapidly improve light conditions, thus creating conditions for the recovery of submerged macrophytes [14–16]. In addition, the change in water depth may also change the interspecific relationships and community structure of submerged macrophytes due to the differences in photosynthetic characteristics, morphological plasticity and reproductive strategies among different plant species [17]. For example, Gao found that the biomass and relative growth rate of *Myriophyllum spicatum* and *Potamogeton malaianus*, canopy-forming plants, increased with increasing water depth (from 2.6 to 3.8 m), while the dominance of *Hydrilla verticillata,* which is an erect plant, decreased significantly [18].

*Vallisneria natans* and *H. verticillata* are common submerged macrophyte species in shallow lakes in China, and are also widely used as pioneer plant species in lake restoration. *V. natans* can tolerate a certain degree of low light in turbid waters due to its low light-compensation point, while *H. verticillata* tends to dominate in high-transparency waters due to its high light-compensation point [18]. There may be solid interspecific competition between the two submerged macrophytes in natural water bodies. Most of the previous studies focused on the response of a single submerged macrophyte to water level/depth or light at the population level [18,19], but the effect of water depth on the competitive abilities of two submerged macrophytes has rarely been reported for water bodies where the two species coexist. In this study, outdoor mesocosm experiments were conducted to investigate the effects of water depth on the growth of the two submerged macrophytes. It is assumed that an increase in water depth will decrease the biomass of both submerged macrophytes but increase the domination of *V. natans* due to its low light-compensation point.

## 2. Materials and Methods

### 2.1. Experimental Design

The outdoor mesocosm experiment was conducted in an experimental pool (3 m in length, 3 m in width, and 2.5 m in height) at Taihu Laboratory for Lake Ecosystem Research (31°30′ N, 120°30′ E), located in Meiliang Bay on the northern edge of Lake Taihu, China. The mesocosm experiment lasted 49 days, from 22 August to 10 October 2018. Young similar-sized seedlings of *V. natans* and *H. verticillata* with good growth and uniform color were collected from a nearby river one week before the experiment. They were precultivated with lake water and lake sediment in plastic bins outdoors. Before the experiment began, the pool was cleaned and filled with lake water, which was taken from Meiliang Bay of Taihu Lake, filtered through a 64 μm plankton net to remove planktonic crustaceans, and slowly injected.

On the day of the experiment, the mixed lake sediment (10 cm) was placed into plastic containers (height 16 cm, the diameter of upper and lower bottom 41 cm and 33 cm, respectively), and then 15 uniform *V. natans* (each with approximately 5–6 bright green leaves, a length of 15 ± 0.5 cm) and *H. verticillata* (a length of 15 ± 0.5 cm) plants were transplanted into each mesocosm. Sediment (TN: 2.04 ± 0.06 mg g$^{-1}$; TP: 0.32 ± 0.01 mg g$^{-1}$) was also collected from Meiliang Bay of Taihu Lake, filtered with a screen (mesh size: 1.7 mm) to remove snails and shellfish, mixed, and added. The plastic mesocosm was quickly added to the experimental pool. The initial total wet weight of *V. natans* was 21.82 ± 0.30 g, and the density was approximately 170 g m$^{-2}$, while the initial total wet weight of *H. verticillata* was 8.89 ± 0.17 g, and the density was approximately 68 g m$^{-2}$. The density of the two submerged macrophytes was in the range of the summer submerged macrophyte density in West Taihu Lake [20]. Three water depth gradients were set up in the experiment, which were named the shallow water depth group (50 cm), intermediate water depth group (150 cm), and deep water depth group (250 cm). There were four replicates in each gradient



and 12 experimental units in total. The three water depths corresponded well with the range of annual fluctuations in some shallow lakes in the middle and lower reaches of the Yangtze River.

*2.2. Sampling and Analytical Methods*

Water samples were collected weekly during the experiment. Prior to water sampling, the light intensity in each mesocosm was measured in situ at four depths (0 cm, 50 cm, 150 cm, and 250 cm) using an underwater digital luxmeter (ZDS-10W, Yueci Electronic Technology Co., Ltd., Shanghai, China).

Water samples (5 L) were also collected weekly from each mesocosm using an acrylic tube sampler. The samples were analyzed for nutrient contents and chlorophyll *a* (Chl-*a*). Unfiltered water samples were analyzed for total phosphorus (TP) and total nitrogen (TN) by colorimetry after digestion with $K_2S_2O_8$ and NaOH solutions. The same methods were used for analyzing total dissolved phosphorus (TDP) and total dissolved nitrogen (TDN), with the exception of the samples being filtered before analysis [21] (Jin and Tu 1990). Chl-*a* was determined from filtered matter retained on a glass microfiber filter (Whatman GF/C International Ltd., Maidstone, England) and extracted in a 90% acetone/water solution over 24 h, after which the concentrations were measured using a spectrophotometer (UV-2450, Shimadzu Co., Ltd., Kyoto, Japan) [22].

All samples of *V. natans* and *H. verticillata* were collected at the end of the experiment (10 October) and thoroughly rinsed with running water to estimate the total wet biomass and the mean number of plants in each mesocosm. To ensure that the water content of the wet biomass was consistent across samples and time, before weighing, the washed plants were allowed to drain on the absorbent paper for 10 min. Afterward, the plant materials were gently pressed on the paper until no water appeared on the paper. Then, the biomass (wet weight/area) and relative growth rate (RGR) were calculated; the RGR was estimated using the following equation:

$$\text{RGR (mg g}^{-1}\text{ d}^{-1}) = \ln (W_f/W_i)/D, \tag{1}$$

where $W_f$ (g) and $W_i$ (g) are the final and initial total wet weights of plants in each mesocosm, respectively, and D is days [23].

In addition, five *V. natans* and five *H. verticillata* were randomly selected from each mesocosm to measure the plant numbers and plant height of *V. natans* and the branch number and plant height of *H. verticillata*. Finally, the average value of five plants was used to represent the morphological data of plants in the experimental mesocosm.

One-way analysis of variance (one-way ANOVA) was used to reveal the effects of water depth on biomass and morphological indicators of *V. natans* and *H. verticillata* at the end of the experiment. Data were $\log_{10}x$ transformed to meet the requirements of normal distribution and homogeneity of variance. All comparisons were conducted using the statistical package SPSS, version 19.0, and all figures were plotted by Prism 7.0.

## 3. Results

*3.1. Water Physical and Chemical Parameters*

Underwater light conditions decreased with increasing in water depth (Table 1). The effects of water depth on nutrient concentrations and chlorophyll *a* concentrations (Chl-*a*) were apparently minor in our experiment (Table 1).

**Table 1.** Changes in light (light intensity of target depth to water surface), total nitrogen (TN), total phosphorus (TP), total dissolved nitrogen (TDN), total dissolved phosphorus (TDP), and chlorophyll *a* (Chl-*a*) concentrations during the experiment.

| | Shallow | | | Intermediate | | | Deep | | |
|---|---|---|---|---|---|---|---|---|---|
| | **Max** | **Min** | **Average** | **Max** | **Min** | **Average** | **Max** | **Min** | **Average** |
| Light (target depth: surface) | 0.437 | 0.181 | 0.304 | 0.132 | 0.029 | 0.075 | 0.048 | 0.005 | 0.023 |
| TN (mg L$^{-1}$) | 1.60 | 1.04 | 1.26 | 1.36 | 1.06 | 1.21 | 1.41 | 1.04 | 1.21 |
| TP (µg L$^{-1}$) | 70 | 42 | 56 | 67 | 44 | 57 | 76 | 47 | 60 |
| TDN (mg L$^{-1}$) | 0.96 | 0.78 | 0.87 | 0.95 | 0.79 | 0.88 | 0.94 | 0.78 | 0.87 |
| TDP (µg L$^{-1}$) | 37 | 20 | 28 | 37 | 21 | 28 | 39 | 20 | 28 |
| Chl-*a* (µg L$^{-1}$) | 28 | 5 | 17 | 25 | 6 | 15 | 27 | 6 | 17 |

*3.2. Growth of Submerged Macrophytes*

Along with the increase in water depth, we found significant declining trends in biomass, RGR, and plant numbers of *V. natans* (Table 2; Figure 1a–c). Water depth also significantly affected the height of *V. natans* (Table 2; Figure 1d). The height of *V. natans* was significantly higher in the intermediate water depth group than in the shallow and deep water depth groups. The latter resulted in the lowest height (Figure 1d).

**Table 2.** One-way ANOVA results of the growth and morphological indices of both submerged macrophytes among different water depths.

| | Variables | F | Df | *p* |
|---|---|---|---|---|
| *V. natans* | Biomass | 134.48 | 2.00 | <0.001 |
| | RGR | 28.76 | 2.00 | <0.001 |
| | Numbers | 88.94 | 2.00 | <0.001 |
| | Height | 124.57 | 2.00 | <0.001 |
| *H. verticillata* | Biomass | 256.56 | 2.00 | <0.001 |
| | RGR | 31.93 | 2.00 | <0.001 |
| | Numbers | 657.42 | 2.00 | <0.001 |
| | Height | 224.94 | 2.00 | <0.001 |
| Total | Total biomass | 505.88 | 2.00 | <0.001 |
| | *V. natans*: *H. verticillata* | 35.21 | 2.00 | <0.001 |

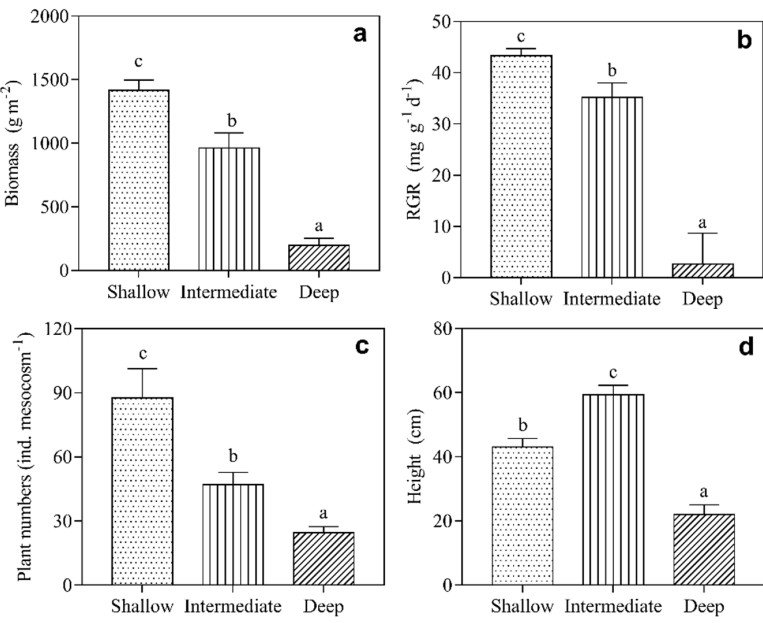

**Figure 1.** Comparisons of (**a**) biomass (g m$^{-2}$), (**b**) RGR (mg g$^{-1}$ d$^{-1}$), (**c**) plant numbers (ind. mesocosm$^{-1}$), and (**d**) plant height (cm) of *V. natans* among three water depth groups. Values represent mean ± SD (n = 4). Means with different letters are significantly different (*p* < 0.05).

Water depth significantly affected the biomass, RGR, and plant branch numbers of *H. verticillata* (Table 2; Figure 2a–c). The biomass, RGR, and plant branch numbers were notably lower in the deep water group than in the shallow and intermediate water groups. However, there was no significant difference between shallow and intermediate water depths (Table 2; Figure 2a–c). In addition, water depth also notably affected the height of *H. verticillata*. As the water depth increased, the height of *H. verticillata* increased (Table 2; Figure 2d).

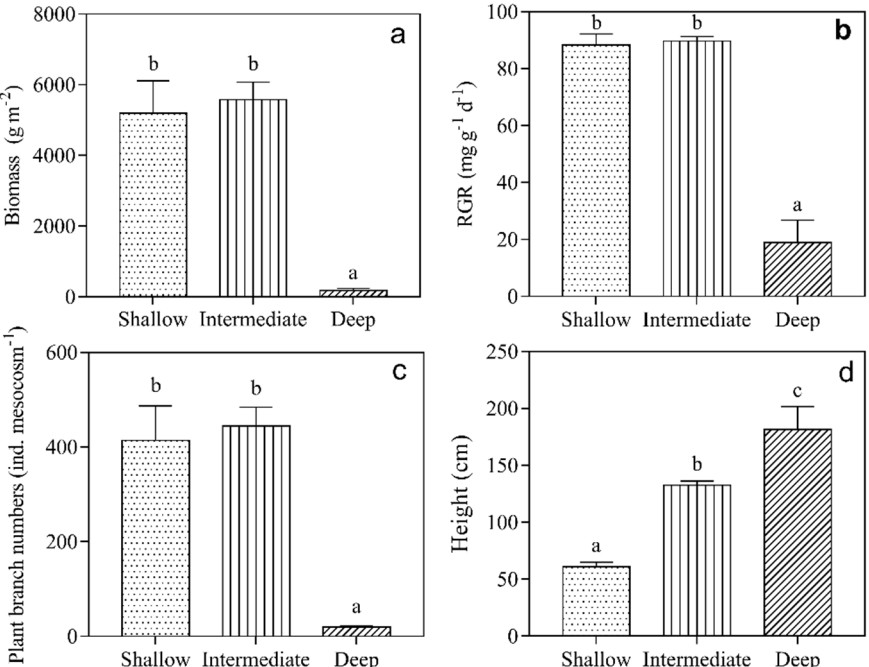

**Figure 2.** Comparisons of (**a**) biomass (g m$^{-2}$), (**b**) RGR (mg g$^{-1}$ d$^{-1}$), (**c**) plant branch numbers (ind. mesocosm$^{-1}$), and (**d**) plant height (cm) of *H. verticillata* among three water depth groups. Values represent mean $\pm$ SD (n = 4). Means with different letters are significantly different ($p < 0.05$).

Water depth also prominently affected the total biomass and the biomass ratio of *V. natans* to *H. verticillata* (Table 2; Figure 3). The total biomass of *V. natans* to *H. verticillata* was significantly lower in deep water than in the shallow and intermediate water depth groups. However, there were no significant differences between the shallow and intermediate water groups (Table 2; Figure 3). At the same time, the biomass ratio of *V. natans* to *H. verticillata* in deep water was significantly higher than that at shallow and intermediate water depths. There were no significant differences between the shallow and intermediate water depth groups (Table 2; Figure 3).

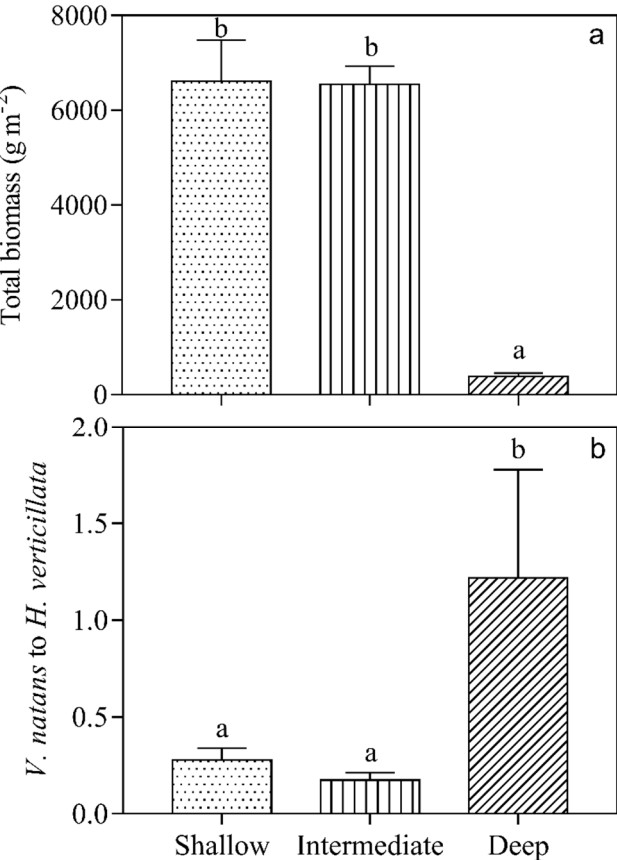

**Figure 3.** Comparisons of (**a**) total macrophyte biomass (g m$^{-2}$) and (**b**) biomass ratio of *V. natans* to *H. verticillata* among the three water depth groups. Values represent mean ± SD (n = 4). Means with different letters are significantly different ($p < 0.05$).

## 4. Discussion

By mixing plantings of *V. natans* and *H. verticillata*, our study aimed to explore the effect of water level fluctuations on the growth of these two submerged macrophytes and their competition. As expected, we found that increased water depth (from 50 cm to 250 cm) significantly reduced the total biomass of submerged macrophytes due to changes in light penetration, which were in line with most mesocosm experiments and field investigations [24–26].

Laboratory studies have found that *V. natans* has a low light-compensation point relative to other submerged macrophyte species, which causes *V. natans* to often grow at greater depths in lakes [27]. In our study, we found that, with the increase in water depth, the biomass and RGR of *V. natans* plants showed a decreasing trend (Figure 1a–c). Similar results were also found in mesocosm experiments by Li et al. (2019), Diao et al. (2017), and Fu et al. (2012), indicating that increased water depths would significantly suppress *V. natans* growth in spite of its low light-compensation point [24,28,29]. In our study, we found that plant numbers of *V. natans* decreased significantly with increasing water depth, while plant height was higher in the intermediate water depth treatment than in the shallow water depth treatment (Figure 1c,d). This result is similar that of Xiao et al. (2007), who found that, with good light conditions, *V. natans* occupied more space by increasing the length of stolons for clonal reproduction. Under poor light conditions, *V. natans* obtained sufficient light by vertical expansion, such as increasing plant height, while horizontal proliferation decreased [30]. However, in our study, both plant numbers and plant height were lowest in the deep water, suggesting that both horizontal proliferation and vertical growth were inhibited when light was extremely low. In this case, *V. natans* may allocate more resources to the below-ground part, e.g., roots and rhizome [9,31].

As an erect plant, *H. verticillata* has a higher light-compensation point than *V. natans*, and is thought to be less light tolerant in lakes [27]. Additionally, our results support this view, as we found that both biomass and RGR in deep water were the lowest among three water depth groups (Figure 2). In our study, light intensity in deep water was generally below 1000 lx and low light intensity limited the growth of *H. verticillata*. Previous studies suggested that *H. verticillata* adapted to low-light conditions by increasing their height and to high-light conditions by forming a canopy, reducing the damage to the lower part of the plant [32]. This phenomenon was also observed in our study; that is, in the shallow and intermediate water depth groups, *H. verticillata* formed a relatively closed canopy by reducing plant height and increasing the number of branches. In the deep water depth group, it showed increased plant height and reduced numbers of ramets to cope with the low light intensity. Mesocosm experiments conducted by Wu et al. (2011) also showed that within the range of 0.5–2.0 m, the plant height of *H. verticillata* was negatively and linearly related to water depth, similar to the results of this study [33].

Our results also show that the changes in water depth affected the competition patterns of the two submerged macrophytes, since the proportion of *V. natans* among the total biomass of the submerged macrophytes increased with increasing water depth, while *H. verticillata* showed the opposite trend (Figure 3). The results are consistent with the photosynthetic characteristics and water depth distribution of the two plants in shallow lakes [34–36], suggesting that increased light penetration would favor the dominance of erect plant species. This may have implications in lake restoration, where water level is often adjusted to increase the light conditions for the colonization of submerged macrophytes [37]. Our results suggest that, although this change in water level/depth may increase the total biomass of submerged macrophytes, it may also cause a high dominance of erect-type *H. verticillata*, while key populations such as *Vallisneria spinulosa* may be difficult to develop [15]. Therefore, the photosynthetic characteristics of submerged macrophytes should be fully considered in the context of water level regulations to ensure the development of the whole community and maximize the competitive advantages of target plants.

**Author Contributions:** Q.L., H.H. and K.L. designed the study; Q.L., Y.H. and K.C. conducted the sampling; Q.L., Y.H., X.H., K.L. and H.H. conducted the data analyses and wrote the paper. All authors have read and agreed to the published version of the manuscript.

**Funding:** This research was funded by the National Science Foundation of China (No. 31930074; 31971473).

**Institutional Review Board Statement:** Not applicable.

**Informed Consent Statement:** Not applicable.

**Data Availability Statement:** Data are presented in the text.

**Acknowledgments:** We are grateful to Jingchen Xue and Rongshu Qian for samples analysis. This study was financially supported by the National Science Foundation of China (No. 31930074; 31971473).

**Conflicts of Interest:** The authors declare no conflict of interest.

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
