# Peer review of "Effects of Water Depth on the Growth of the Submerged Macrophytes Vallisneria natans and Hydrilla verticillata: Implications for Water Level Management"

_water, doi:10.3390/w13182590_

Round 1

Reviewer 1 Report

Please, find the comments in the manuscript.

Author Response

-Line 29: Keywords: Do not repeat word from the title in Keywords.

Our response: We now changed keywords into Lake restoration; Morphological response; Interspecific competition, Light attenuation” to avoid repetition (Line 28).

-Line 33: “large aquatic plants”: “large aquatic plants which encompass vascular plants, bryophytes and charophytes...”

Our response: We have deleted this sentence to avoid misunderstanding.

-Line 34: “shallow lakes”: “large aquatic plants which encompass vascular plants, bryophytes and charophytes...”

Our response: Thank you for your kind advice. We have replaced “shallow lakes” with “freshwater lakes” in the revised manuscript (Line 31).

-Line 43: “submerged”: “It is not necessary to repeat "submerged" since in the first sentence macrophytes are defined as underwater plants.”

Our response: As we deleted the first sentence involved the definition of submerged macrophytes, we kept “submerged” here (Line 39-40).

-Line 62: “H. verticillate”: “Use full generic name in the first quotation of certain plant species. I.e. Hydrilla verticillata.”

Our response: We have changed “H. verticillate” into “Hydrilla verticillate” (Line 59).

-Line 64: “V. natans”: “Same comment as previous. Use the full name, Vallisneria natans.”

Our response: We have changed “V. natans” into “Vallisneria natans” in the revised manuscript (Line 61).

-Line 83: “August 22nd to October 10th”: “Please, add the year”

Our response: We have added year (2018) in the revised manuscript (Line 80).

-Line 84: “the submerged macrophytes”: “"the submerged macrophytes" is not necessary. It is already defined that these two species are submerged macrophytes”

Our response: We have deleted “the submerged macrophytes” here in the revised manuscript (Line 81).

-Line 85: “were collected”: “Write where they were collected. In natural habitats? Which one(s)?”

Our response: Young similar-sized seedlings of V. natans and H. verticillata with good growth and uniform color were collected from rivers near the Lake Taihu. We have now stated it clearly in the revised manuscript (Line 82).

-Line 108: “at weekly”: “weekly, omit “at”

Our response: We have deleted “at” in the revised manuscript (Line 105).

-Line 114: “TP and TN”: “Perhaps is better to explain here the abbreviations, e.g., "total phosphorus (TP) and total nitrogen (TN)."

Our response: We have changed “TP and TN” into “total phosphorus (TP) and total nitrogen (TN)”in the revised manuscript (Line 111).

-Line 122: “All V. natans and H. verticillate”: “All samples of V. natans and...”

Our response: We have changed “All V. natans and H. verticillate” to “All samples of V. natans and H. verticillate” in the revised manuscript (Line 119).

-Line 129-130: “Ln”: “"ln" instead "Ln"”

Our response: We have changed “Ln” into “ln” in the revised manuscript (Line 126-127).

-Line 138: “If necessary, the data were log10x transformed”: “My recommendation is to make log transformation on all data. It is not so good to have some data transformed and some not”

Our response: Now, we applied log transformation to all data. We have stated this clearly in the Method (Line 135), and modified results accordingly.

-Line 232: “H. verticillate”: “H. verticillata

Our response: Corrected, thanks (Line 229).

Author Response

Making use of mesocosm experiments, the paper deals with the effects of water-level fluctuations (WLFs) on the growth and competition between two macrophyte species. Since lakes and rivers with regulated level are continuously increasing due to augmented water demand and climate change, the ecological effects of WLFs are a very timely topic. My main concern was the poor language of important parts of the ms (title, Abstract, Discussion). Since the other parts of the ms were more or less OK from this point of view, to help the authors fix this, I directly edited the most important parts (see my referee report, changes highlighted in yellow).

I think that the paper can be accepted after carrying out these minor revisions:

Our response: We would like to thank you for your careful reading, helpful comments, and constructive suggestions, which greatly improved our manuscript. We have carefully considered all comments from the reviewers and revised our manuscript accordingly.

Specific comments:

- Line 42: Please replace “The water level” with “Water level”.

Our response: We have now replaced “The water level” with “Water level” (Line 39).

- Line 62: Please replace “which are erect plants” with “which is an erect plant”.

Our response: We have replaced “which are erect plants” with “which is an erect plant” in the revised manuscript (Line 59).

- Line 65: Please replace “and also widely” with “and are also widely”.

Our response: We have added “are” in the revised manuscript (Line 62).

- Line 66: Correct to “light-compensation”.

Our response: We have changed “light compensation” into “light-compensation” in the revised manuscript (Line 64).

- Line 76: Correct to “light-compensation”.

Our response: We have corrected “light compensation” to “light-compensation” in the revised manuscript (Line 65).

- Line 88: Please replace “64-micron” with “64-μm”.

Our response: We have replaced “64-micron” with “64-μm” in the revised manuscript (Line 85).

- Line 95: “10-mesh screen”. What is this?

Our response: It is a screen, with a mesh size of 1.7 mm. We now changed “a 10-mesh screen” to “a screen (mesh size:1.7 mm)” in the revised manuscript (Line 92).

- Line 102: Please replace “low water” with “shallow water”.

Our response: We have changed “low” into “shallow” throughout the manuscript (Line 99).

- Line 104: Correct to “The three water depths corresponded well”.

Our response: We have corrected “Three water depths were corresponded well” to “The three water depths corresponded well” in the revised manuscript (Line 101).

- Line 108: Correct to “collected weekly”.

Our response: We have corrected “collected at weekly” to “collected weekly” in the revised manuscript (Line 105).

- Line 120: Correct to “measured using a spectrophotometer”.

Our response: We have corrected “measured by using a spectrophotometer” to “measured using a spectrophotometer” in the revised manuscript (Line 117).

- Line 127: Correct to “pressed on the paper until no water appeared. Then,…”.

Our response: We have changed “to” into “on” in the revised manuscript (Line 124).

- Line 143: Correct to “physical and chemical”. Strictly speaking only electrical conductivity is a physicochemical parameter.

Our response: We have now corrected to “physical and chemical” in the revised manuscript (Line 139).

- Line 144: Correct to “physical and chemical”.

Our response: Follow the suggestion of reviewer, we have now deleted this sentence.

- Line 146: Correct to “depth, light obviously”.

Our response: In the revised manuscript, we have changed this sentence into “Underwater light conditions decreased with increasing in water depth (Table 1).” (Line 140).

- Line 147: Correct to “(Chl-a) were apparently minor”.

Our response: We have corrected to “(Chl-a) were apparently minor” in the revised manuscript (Line 141-142).

- Line 148: Period (dot) at the end of the sentence missing.

Our response: We have now added dot at the end of the sentence (Line 142).

- Line 148: Insert comma after “(TDP)”.

Our response: Corrected (Line 144).

- Table 1: Headings: “Deep”, “Average”… etc. on the right in bold. Suggest reporting TDN and TDP in μg L−1. Chl-a: No decimal digits for values in μg L−1.

Our response: Sorry for the wrong presentation of data. We have now revised it accordingly (Table 1).

- Lines 153-154: Correct to “trends in biomass”.

Our response: Corrected (Line 147-148).

- Line 154: Insert comma after “RGR”.

Our response: Corrected (Line 148).

- Line 172: Correct to “total biomass ratio of”.

Our response: We have corrected to “biomass ratio of V. natans to H. verticillata” in the revised manuscript (Line 167).

- Line 173: Correct to “total biomass ratio of”.

Our response: Corrected (Line 168).

- Lines 173-174: Correct to “lower in deep water than in shallow and”.

Our response: We have corrected to “lower in deep water than in shallow and” in the revised manuscript (Line 169).

- Line 176: Correct to “in deep water was significantly”.

Our response: We have corrected to “in deep water was significantly” in the revised manuscript (Line 172).

- Line 177: Correct to “water depths. There”.

Our response: We have corrected to “water depths. There” in the revised manuscript (Line 173).

Comments to the Author:

I edited important parts of the ms (title, Abstract, Discussion) that were most badly in need of linguistic polishing.

Our response: We would like to thank you for your careful reading, helpful comments, and constructive suggestions, which greatly improved our manuscript. We have carefully considered all comments from the reviewers and revised our manuscript accordingly.

Specific comments:

Title

- replace “in” with “on”

Our response: We have replaced “in” with “on” in the revised manuscript (Line 2).

- replace “Natans” with “natans

Our response: We have replaced “Natans” with “natans” in the revised manuscript (Line 3).

- replace “Verticillata” with “verticillate

Our response: We have replaced “Verticillata” with “verticillate” in the revised manuscript (Line 3).

Abstract

- replace “The water” with “Water”

Our response: We have replaced “The water” with “Water” in the revised manuscript (Line 11).

- insert “the” before “erect plant”

Our response: We have inserted “the” before “erect plant” in the revised manuscript (Line 12).

- replace “fluctuations in water levels” with “water-level fluctuations”

Our response: We have replaced “fluctuations in water levels” with “water-level fluctuations” in the revised manuscript (Line 14).

- replace “remains to be studied” with “was still unknown”

Our response: We have replaced “remains to be studied” with “was still unknown” in the revised manuscript (Line 14).

- insert comma after “150 cm”

Our response: We have inserted comma after “150 cm” in the revised manuscript (Line 15).

- replace “The results” with “Results”

Our response: We have replaced “The results” with “Results” in the revised manuscript (Line 17).

- insert comma after “that”

Our response: We have inserted comma after “that” in the revised manuscript (Line 17).

- replace “low water depth group” with “shallow water condition”

Our response: We have replaced “low water depth group” with “shallow water condition” in the revised manuscript (Line 17).

- replace “were” with “was”

Our response: We have replaced “were” with “was” in the revised manuscript (Line 18).

- replace “under deep water depth” with “in deep water”

Our response: We have replaced “under deep water depth” with “in deep water” in the revised manuscript (Line 18).

- Correct to “Moreover, the ratio of biomass of V. natans to H. verticillata gradually increased with increasing water depth, indicating that deep water enhanced the competitive advantage of V. natans over H.verticillata.”.

Our response: Corrected (Line 19-21).

- replace “The morphological” with “Morphological”

Our response: We have replaced “The morphological” with “Morphological” in the revised manuscript (Line 21).

- replace “theirs” with “its”

Our response: We have replaced “theirs” with “its” in the revised manuscript (Line 23).

- change “with” into “at”

Our response: We have changed “with” to “at in the revised manuscript (Line 23).

- replace “the” with “to”

Our response: We have replaced “the” with “to” in the revised manuscript (Line 23).

- replace “Similar” with “A similar”

Our response: We have replaced “Similar” with “A similar” in the revised manuscript (Line 24).

- insert comma after “V. natans

Our response: We have inserted comma after “V. natans” in the revised manuscript (Line 24).

- replace “water level” with “water-level”

Our response: We have replaced “water level” with “water-level” in the revised manuscript (Line 26).

- insert “such as” before “V. natans

Our response: We have inserted “such as” before “V. natans” in the revised manuscript (Line 27).

Discussion

- replace “both” with “these two”

Our response: We have replaced “both” with “these two” in the revised manuscript (Line 183).

- replace “to the changes of” with “to changes in”

Our response: We have replaced “to the changes of” with “to changes in” in the revised manuscript (Line 185).

-replace “deeper” with “at greater depths”

Our response: We have replaced “deeper” with “at greater depths” in the revised manuscript (Line 189-190).

- insert comma after “that”

Our response: We have inserted comma after “that” in the revised manuscript (Line 190).

- replace “of” with “in”

Our response: We have replaced “of” with “in” in the revised manuscript (Line 190).

- correct to “indicating”

Our response: Corrected (Line 193).

- correct to “suppress”

Our response: Corrected (Line 193).

- insert “that” after “found”

Our response: We have inserted “that” after “found” in the revised manuscript (Line 195).

- insert “that,” after “found”

Our response: We have inserted “that,” after “found” in the revised manuscript (Line 197).

- change “With” into “Under”

Our response: We have changed “With” to “Under” in the revised manuscript (Line 199).

- delete “the” after “both”

Our response: We have deleted “the” after “both” in the revised manuscript (Line 201).

- replace “the lowest in the deep water depth group” with “lowest in deep water”

Our response: We have replaced “the lowest in the deep water depth group” with “lowest in deep water” in the revised manuscript (Line 201-202).

- replace “suggested” with “suggesting”

Our response: We have replaced “suggested” with “suggesting” in the revised manuscript (Line 202).

-replace “the light conditions were” with “light was”

Our response: We have replaced “the light conditions were” with “light was” in the revised manuscript (Line 203).

- delete “choose to”

Our response: We have deleted “choose to” in the revised manuscript (Line 203).

- replace “a” with “an”

Our response: We have replaced “a” with “an” in the revised manuscript (Line 205).

- delete “that of”

Our response: We have deleted “that of” in the revised manuscript (Line 205).

- insert “is” before “thought”

Our response: We have inserted “is” before “thought” in the revised manuscript (Line 206).

- Correct to “Also our results supported this view as we found that both biomass and RGR in deep water were the lowest among the three water depth groups (Figure 2)”.

Our response: Corrected (Line 206-208).

- Correct to “In our study, light intensity in deep water was generally below 1000 lx, and low light intensity limited the growth of H. verticillate.”.

Our response: Corrected (Line 208-209).

- Corrected to “Previous studies suggested that H. verticillata adapted to low and high light conditions by increasing its plant height and formed a canopy (reduced the damage to plants caused by excessive light) in the upper water body [32].”.

Our response: We have changed the sentence into “Previous studies suggest that H. verticillata adapts to low light conditions by increasing their height and to high light conditions by forming a canopy, reducing the damage to the lower part of the plant [32].” in the revised manuscript (Line 209-212).

- replace “experiment” with “experiments”

Our response: We have replaced “experiment” with “experiments” in the revised manuscript (Line 216).

- replace “linear” with “linearly”

Our response: We have replaced “linear” with “linearly” in the revised manuscript (Line 217).

- replace “macrophytes as” with “macrophytes, since”

Our response: We have replaced “macrophytes as” with “macrophytes, since” in the revised manuscript (Line 220).

- replace “suggest” with “suggesting”

Our response: We have replaced “suggest” with “suggesting” (Line 224).

- Correct to “This may have implications in lake restoration, where water level is often adjusted to increase the light conditions for the colonization of submerged macrophytes [37].”.

Our response: Corrected (Line 225-227).

- insert comma after “that”

Our response: We have inserted comma after “that” in the revised manuscript (Line 227).

- replace “but it may” with “it may also”

Our response: We have replaced “but it may” with “it may also” in the revised manuscript (Line 228).

- correct “H. verticillate” to “H. verticillata

Our response: Corrected (Line 229).

- delete “the” before “key”

Our response: We have deleted “the” before “key” in the revised manuscript (Line 229).

- replace “V. spinulosa” with “Vallisneria spinulosa

Our response: We have replaced “V. spinulosa” with “Vallisneria spinulosa” in the revised manuscript (Line 229).